# ZnO Treatment on Mechanical Behavior of Polyethylene/Yellow Birch Fiber Composites When Exposed to Fungal Wood Rot

**DOI:** 10.3390/polym15183664

**Published:** 2023-09-06

**Authors:** Kodjovi Kekeli Agbozouhoue, Demagna Koffi, Fouad Erchiqui, Simon Barnabé

**Affiliations:** 1Innovations Institute in Ecomaterials, Ecoproducts and Ecoenergy (I2E3), Université du Québec à Trois-Rivières (UQTR), C.P. 500, Trois-Rivières, QC G9A 5H7, Canada; kekeli.agbozouhoue@uqtr.ca (K.K.A.); simon.barnabe@uqtr.ca (S.B.); 2Laboratory of Bioplastic and nanomaterials, School of Engineering, Université du Québec en Abitibi Témiscamingue (UQAT), 445 Bd de l’Université, Rouyn-Noranda, QC J9X 5E4, Canada; fouad.erchiqui@uqat.ca

**Keywords:** *Trametes versicolor*, *Gloephyllum trabeum*, mycelium, Young’s modulus, zinc oxide, composite, impact energy

## Abstract

Wood plastic composite (WPC) usage and demand have increased because of its interesting chemical and mechanical properties compared to other plastic materials. However, there is a possibility of structural and mechanical changes to the material when exposed to the external environment; most research on wood plastic is performed on the material with elevated fiber content (40–70%). Therefore, more research needs to be performed regarding these issues, especially when the fiber content of the WPC is low. In this study, composite materials composed of high-density polyethylene (HDPE) reinforced with yellow birch fibers (20 and 30%) were made by injection molding. The fibers were treated with dissolved zinc oxide (ZnO) powder in sodium oxide (NaOH) solution, and the fabricated material was exposed to fungal rot. ZnO treatment in this case is different from most studies because ZnO nanoparticles are usually employed. The main reason was to obtain better fixation of ZnO on the fibers. The mechanical properties of the composites were assessed by the tensile and Izod impact tests. The impact energies of the samples fabricated with ZnO-treated fibers and exposed to *Gloephyllum trabeum* and *Trametes versicolor* decreased, when compared to samples fabricated with ZnO-nontreated fibers. The mechanical properties of the samples composed of ZnO-treated fibers and exposed to rot decreased, which were reported by a decreased Young’s modulus and impact energies. The usage of ZnO treatment prevented mycelium proliferation, which was nonexistent on the samples. It has been noted that the decrease in mechanical properties of the treated samples was because of the action of NaOH used to dissolve the ZnO powder.

## 1. Introduction

Biocomposites are composite materials with reinforcements, such as wood fibers. The development of these materials based on natural fibers has many advantages due to their biodegradability and interesting mechanical properties [1,2,3,4,5].

Natural fibers can be obtained from natural sources, like animals, plants, and minerals [6,7]. Vegetal fibers are classified according to their physiological properties. Bast fibers are obtained from flax, hemp, kenaf, jute, etc.; leaf fibers from sisal, curauá, palm, etc.; seed fibers from cotton, soya, kapok, etc.; fruit fibers (coir, luffa, etc.); grass fibers from bamboo, bagasse, etc.; and wood fibers, such as hardwood and softwood (teak wood, birch, etc.) [6,7,8,9,10,11,12,13]. Fibers are chosen depending on their properties, such as mechanical chemical, and physical. It is also important to mention that the availability of a local fiber with interesting properties could be a determining factor for fiber choice. Yellow birch fibers were chosen for our studies because of its abundance in Quebec and because its usage is an alternative for the processing of residuals remaining from the wood exploitation [6].

Wood plastic composite demand is projected to increase considerably between 2016 and 2024 [6,13]. Many sectors (e.g., construction, sporting equipment, and automotive parts) use this material, but their development, for example, from short fibers is limited because of insufficient understanding and knowledge of their mechanical behavior and impact to environmental factors [6,14,15,16].

Three types of fungal wood rot are known: white, brown, and soft rot [17]. The damage to trees and wood can be separated into damage to felled tree, living tree, and stored wood and wood in exterior use and indoor use [18]. Brown-rot fungi often attack structural wood products in North America. The wood attacked by brown-rot fungi becomes brown, and it is degraded by nonenzymatic and enzymatic systems. Cellulolytic enzymes are used in the degradation process by brown-rot fungi, but lignin-degrading enzymes are not involved. Durmaz, S. et al., in their 2016 work, observed that when exposed to fungi, wood mass loss gradually occurs due to the degradation of wood carbohydrates; they also observed that the intensity of lignin bands increased proportionally to exposure time, and this is characteristic of brown-rot fungi. Lignin content remains constant [19]. White-rot fungi decay patterns have different forms. White-rotted wood has a bleached appearance, leaving the wood a spongy mass; it could also appear as selective decay. White-rot fungi have the cellulolytic- and lignin-degrading enzymes and therefore have the potential to degrade the complete wood structure under adequate environmental conditions. Soft-rot fungi attack the lower lignin content of wood and create cavities in the wood’s cell wall. There is not much knowledge of soft-rot degradative enzyme systems, but their degradative mechanisms are reviewed along with the degradative enzymatic and nonenzymatic systems existing in brown- and white-rot fungi [20].

WPCs are known to be more durable structural material than untreated wood because the plastic matrix encloses the wood and diminishes moisture uptake. However, extended WPC exposure will eventually result in fungal growth [21]. In WPCs, plastics are by and large resistant to fungal attack, although a major problem with these materials is that wood in the composite is still susceptible to biological degradation. Many industrialists avoid this risk by fabricating products for interior use with little or no water presence, in order to minimize the risk of fungal attack [22]. Usage of WPC materials in external conditions may affect their performance.

Zinc oxide is widely used in many fields because of its various properties. ZnO and its derivatives suppress the development and growth of fungi and molds. Zinc oxide is added to fungicides to improve their effectiveness. It reacts with silicates (e.g., sodium silicate) to produce zinc silicates, which are water- and fire-resistant materials used as binders in paints, etc. [23]. Zinc oxide has been used in numerous studies on WPCs. It was used in nanoparticle form by various authors, like MRM Farahani et al. [24] and P Marzbani et al. [25].

Therefore, understanding the impact of microorganism exposure and particularly white and brown rots caused by *Trametes versicolor* and *Gloephyllum trabeum*, respectively, on the mechanical properties of HDPE reinforced with ZnO-treated yellow birch fibers is necessary to foresee its possible applications in the external environment, especially when the fiber content is relatively low (20 and 30%). In our studies, the ZnO treatment of fibers was different because the pigment was dissolved in a molar sodium hydroxide solution prior to fiber immersion. ZnO could be efficiently distributed in the fabricated materials because of better homogeneity in fiber structure. 

To achieve these goals, biocomposite samples were fabricated and exposed to fungal rot; the tensile and Izod impact tests were realized, and the results obtained were compared to control samples. 

## 2. Materials and Methods

### 2.1. Fungus

Two strains of fungi were used: *Trametes versicolor* (ATCC 12679), a white-rot fungus, and *Gloephyllum trabeum* (ATCC 11539), a brown-rot fungus. Both strains were provided by SEREX in Rimouski (Amqui, QC, Canada). They were cultivated in potato dextrose agar (PDA) culture media and incubated at 25 °C with a 70% relative humidity for approximately 7 to 10 days (until complete colonization) prior the exposition. 

### 2.2. Birch Fiber Treatment with Zinc Oxide (ZnO)

The zinc oxide ZnO was produced by SIGMA Aldrich (St. Louis, MI, USA). It was a 99.99% trace metal basis with 81.39 g/mol molar weight. It is a white to yellow powder. It was dissolved in a molar NaOH solution. The solution was obtained by adding the ZnO powder and stirring manually; the solution was heated at 80 °C for approximately 2 min until complete dissolution of the ZnO. Around 60 g of ZnO powder was dissolved in approximately 9 L of the solution. The fibers were immersed in the solution until complete absorption. No washing process after immersion was performed, and the treated fibers were dried in stove at 60 °C for 5 days. 

### 2.3. Preparation of Biocomposite Samples

The biocomposite samples were made by mixing the components in a Thermotron mixer (C.W. Brabender; model T-303), (C.W Brabender Instruments Inc., South Hackensack, NJ, USA). A portion of weighted HDPE (High-Density Polyethylene) was melted on the rollers at 170 °C together with Maleic Anhydride-Grafted Polyethylene MAPE (3% of mix weight). The remaining HDPE and the fibers were added and blended at 60 rpm for 7 min. The mix was removed from the roller and remixed for 3 min five times to obtain a uniform composite sheet, which was further granulated, and the test samples were fabricated by injection (Figure 1). Two biocomposite formulations made of HDPE reinforced with 20 and 30 (*w/w*) % fiber were processed by ZHAFIR Plastics Machinery (100-ton Zerus 900 press, Ebermannsdorf, Bavaria, Germany). Figure 1 shows an aspect of the tensile and the Izod impact test specimens, which specifications followed the ASTM D256-10e1 [26] and ISO 527-1 [27] standard. Five replicates of each material were tested (ZnO-treated and -nontreated samples) (Figure 2). Even though in most studies a high fiber content was used (50–70 wt%), we chose low fiber content to analyze fungal attack possibilities. 

### 2.4. Culture Media

Potato dextrose agar was used as culture medium for both strains. After preparation, it was sterilized in an autoclave for 45–60 min at 121 °C followed by a resting period of 30 min under pressure. It was further used to fill the Petri dishes and the other containers through the laminar flow cabinet of the CIPP laboratory, which was previously disinfected using 70% ethanol and UV lamps. An inoculation of the medium by the two fungal strains was performed.

### 2.5. Exposure Tests

The inoculated culture media were totally colonized by the fungal strains for approximately 7 days as attested by a visual inspection performed every 2 days before the samples were directly placed on it. This process, which was carried out according to level 2 biosafety standards, lasted 30 days to avoid working with aged strains. A septuplet of each biocomposite formulation was exposed to each fungal strain. After the incubation period, the samples were characterized by tensile and impact tests.

### 2.6. Impact Tests of Exposed Samples

The impact tests were conducted at the mechanical engineering laboratory of UQTR based on an application of the ASTM D256-10e1 [26] standard on the appropriate samples, using an impact pendulum instrument (Instron CEAST 9050, fabricated by Instron, Norwood, MA, USA) with a 0.5 J hammer. With the pendulum, the resilience, ductile, and/or brittle fracture of the investigated material can be determined. According to ASTM D256-10e1 [26] standard, the breaking energy of the tested material must be between 10 and 90% of the hammer’s capacity used for the Izod impact tests [6,26].

### 2.7. Microscope Observation

Samples were observed with the Scanning Electron Microscope (SEM) (Hitachi, Toronto, ON, Canada) of the electronic microscopy laboratory of I2E3 at the University of Quebec in Trois-Rivières (UQTR).

## 3. Results

### 3.1. Evaluation of Brown and White Growth on Biocomposite Samples

Table 1 gives an evaluation of fungal growth on the ZnO samples. It is expressed in terms of the proportion of the sample surfaces occupied by the fungal growth, following a given nomenclature.

These results suggest that the fungal colonization of the sample surfaces was nonexistent independent of the kind of fungal strain considered and fiber content. The main surfaces of the samples were not covered by the fungal rot in all the cases.

Samples exposed to *G. trabeum* and *T. versicolor* generally showed the same appearance. No trace of colonization was observed; apparently, the hyphae could not propagate. The samples seem to present a lighter color (Figure 3). 

Sample surfaces that were directly in contact with rot during the exposure did not present any mycelium; it means that the fibers contained in the samples were not compromised and decayed by fungal enzymes and decomposition usually started at the surface of the material. It has been demonstrated in other studies that when in contact with fungal rot, WPC mechanical characteristics were affected and mycelium proliferation on samples were visible. ZnO antifungal characteristics are responsible for preventing mycelium growth. Samples after exposure presented a different color; this was due to water absorption, which is a key element for fungal growth. It is important to underline that NaOH may have also compromised the fiber’s chemical and structural characteristics; it provokes the swelling of vegetal fibers. The last statements show that conditions were optimum for fungal growth in normal conditions. 

### 3.2. Impact Test Results

The results are shown in Figure 4. The results in Figure 5 correspond to Koffi et al. 2021 works. All samples tested were fabricated with the same method and using the same parameters in both studies. 

Treated samples and not exposed to rot had an impact energy of 3.418 kJ/m^2^ and 4.076 kJ/m^2^, respectively, with a 20 and 30% fiber content. Nontreated and nonexposed sample impact energies were respectively 5.33 and 5.46 kJ/m^2^ (Figure 5). The impact energies were 4.488 and 4.45 kJ/m^2^ for the 20% ZnO-treated fiber samples and exposed to *G. trabeum* and *T. versicolor*. The impact energies of the samples loaded with 30% ZnO-treated fibers and exposed to *G. trabeum* and *T. versicolor* were 4.452 and 4.282 kJ/m^2^, respectively, while they were 5.676 and 6.538 kJ/m^2^ [6] for the nontreated samples; impact energy decreases of 21.6% and 34.6%. The decrease is due to the alkaline treatment. In our previous works when WPC samples were exposed to *T. versicolor*, an increase in adhesion between fibers and polymeric matrix was noted resulting in an increased impact energy. Schirp et al. 2006 observed similar activity when samples were exposed to the same fungal rot (*T. versicolor).* The decrease in impact energy, which can be related to a decrease in toughness, is a result of a change in the physical and chemical properties of the fibers caused by the NaOH, because in studies realized by authors like MRM Farahani et al., ZnO was used in nanoparticles, and it did not cause such an important decrease in the WPC-treated fibers. 

### 3.3. Tensile Test Results

Young’s modulus and strain were determined (Figure 6), and Figure 7 represent our controls [6].

Tensile test results of all samples and controls are reported in Table 2.

The samples with 20% ZnO-non treated fiber and not exposed to fungal decay had a Young’s modulus (E) of 2.67 ± 0.13 GPa, while the 20% ZnO- treated samples not exposed to rot had a Young’s modulus of 1.42 ± 0.05 Gpa, which shows a decrease of 53% when samples are treated with ZnO. When treated with ZnO and exposed to *T. versicolor,* the samples with 20% fiber had a Young’s modulus of 1.18 ± 0.026 GPa, while the ZnO-nontreated samples exposed to the same decay had a Young’s modulus of 2.30 ± 0.02 Gpa, which represents a decrease of 51% when treated with ZnO. Samples loaded with 30% fiber treated with ZnO and exposed to *T. versicolor* had a Young’s modulus of 0.79 ± 0.02 GPa, while samples with 30% unexposed to *Trametes versicolor* and untreated with ZnO had a modulus of 3.37 ± 0.16 Gpa, meaning a decreased of 76% is observed when samples were treated and exposed. Young’s modulus was 3.22 ± 0.06 GPa for samples with 30% ZnO nontreated fibers and exposed to *T. versicolor*; ZnO-treated samples exposed to the same decay therefore suffered a decrease in Young’s modulus of approximately 75% compared to ZnO-nontreated samples. The results are similar when samples were exposed to the brown-rot fungi. The samples with 20% ZnO-nontreated fibers exposed to rot had a Young’s modulus of 2.36 ± 0.04 GPa, while ZnO-treated fibers exposed to rot had a Young’s modulus of 1.17 ± 0.07 GPa; there is a decrease of 49.57%. The ZnO-nontreated samples with 30% fiber and exposed to *G. trabeum* had a Young’s modulus of 3.42 ± 0.07 GPa, while those treated with ZnO and exposed to the same rot had a Young’s modulus of 0.86 ± 0.01 GPa, a decrease of 74.85%. This is also a consequence of alkaline fiber treatment; in this process, fibers swell and make changes to the structure, dimension, morphology, and mechanical performance as mentioned in ASTM-D 1965 [28]. As the alkaline treatment eliminates celluloses pectin and lignin, which are the principal natural fiber components, the fiber’s tensile strength may decrease [27,28,29,30,31,32,33,34]. A trend of decreasing mechanical properties is observed with increased concentration of NaOH, which is likely in our case.

### 3.4. Electron Microscope Results

There is mycelium presence on sample surface (Figure 8). On the ZnO-treated samples exposed to rot, there was no mycelium presence (Figure 9).

Figure 8 shows the nontreated sample surface with mycelium growth. Zinc oxide avoided the effect of rot during the exposure of the samples; no trace of mycelium was found on any sample at the same time zinc presence was evidenced (Figure 9). EDX analysis corroborates Zn (0.3%) and Na (10.6%) presence despite the encapsulation of wood fiber with HDPE, low fiber content, and limited immersion time of fibers in the NaOH–ZnO solution. The mechanical properties of zinc oxide-treated WPCs were lower than the untreated ones. Alkaline treatment when NaOH concentration is lower than 5% promotes an increase in interfacial strength between the polymeric matrix and lignocellulosic reinforcement (GHASEMI, E. and Farsi, M., 2010), which also improves the mechanical properties of WPCs. Valášek, Petr, et al. 2021 also confirmed that the excessive and long-term action of alkaline treatment of the NaOH solution causes deterioration in the mechanical properties of individual fibers. In our work, NaOH used as a solvent likely provoked deterioration of the treated fibers causing a decrease in mechanical properties. It is also very important to specify sodium zincate (Na_2_ ZnO_2_) presence in the solution. It is formed by sodium cation and zincate anion. This compound, which is a complex salt, has not been studied deeply in the literature, and its implication in antifungal activities is unknown, making it not very important in our study. 

## 4. Conclusions

In this work, ZnO antifungal properties were used for protecting a WPC against fungal attack. The yellow birch fibers were treated with a ZnO–NaOH solution before WPC fabrication. WPCs were exposed to *Trametes versicolor* and *Gloephyllum trabeum*, white- and brown-rot fungi, respectively. NaOH affected fibers provoking a decrease in mechanical properties. Despite taking advantage of ZnO antifungal characteristics, its dissolution in NaOH was a disadvantage for the WPC mechanical properties. The samples with 20% ZnO-treated fibers not exposed to fungal decay had a Young’s modulus (E) of 2.67 ± 0.13 GPa, while samples with 20% ZnO-treated fibers not exposed to fungal decay had a Young’s modulus of 1.42 ± 0.05 GPa representing a 53% decrease. The results show the efficiency of ZnO, but its use would be optimized in nanoparticle form without compromising WPC performance. 

## Figures and Tables

**Figure 1 polymers-15-03664-f001:**
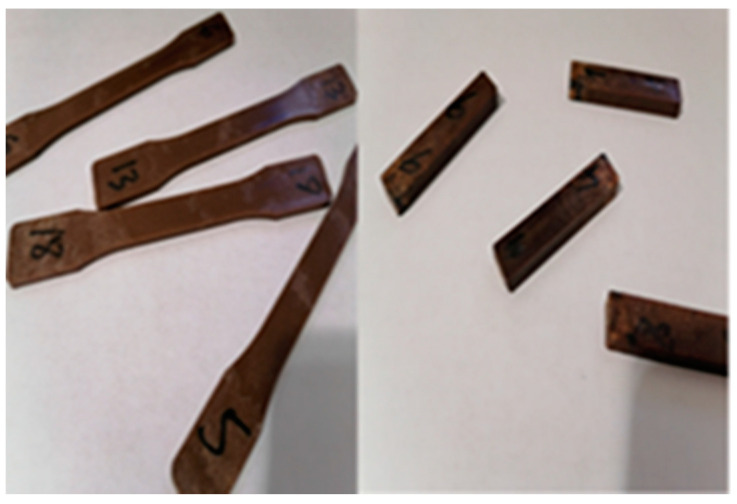
Tensile test samples are the largest and Charpy samples the smallest.

**Figure 2 polymers-15-03664-f002:**
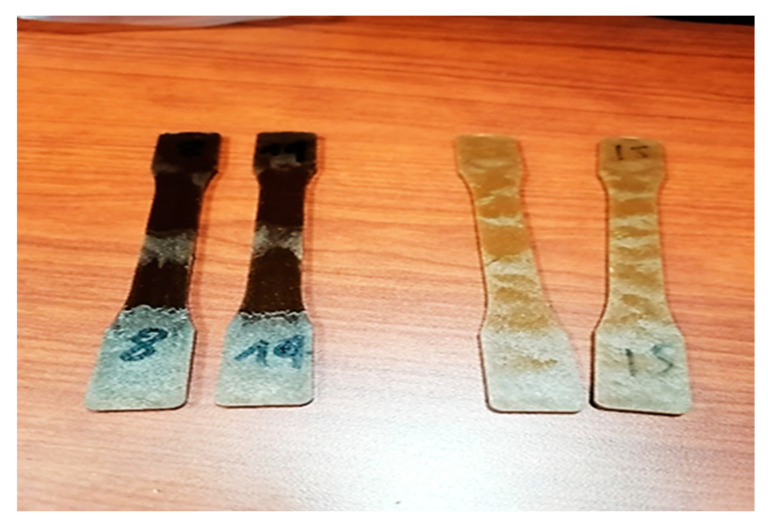
ZnO-nontreated tensile test samples (darkest) and ZnO-treated samples.

**Figure 3 polymers-15-03664-f003:**
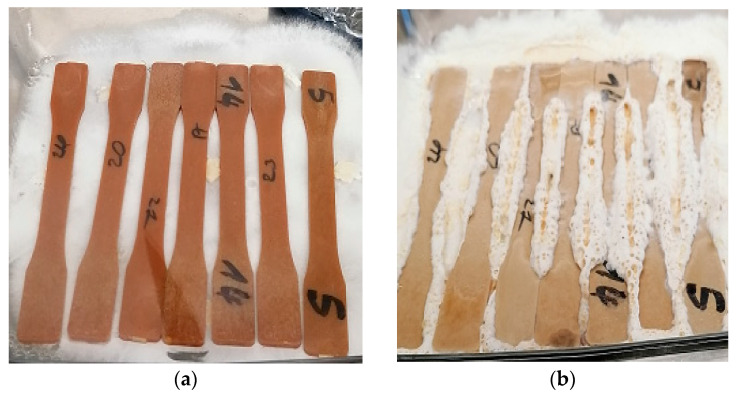
Samples exposed at day 1 of exposure (**a**) and day 30 of exposure (**b**).

**Figure 4 polymers-15-03664-f004:**
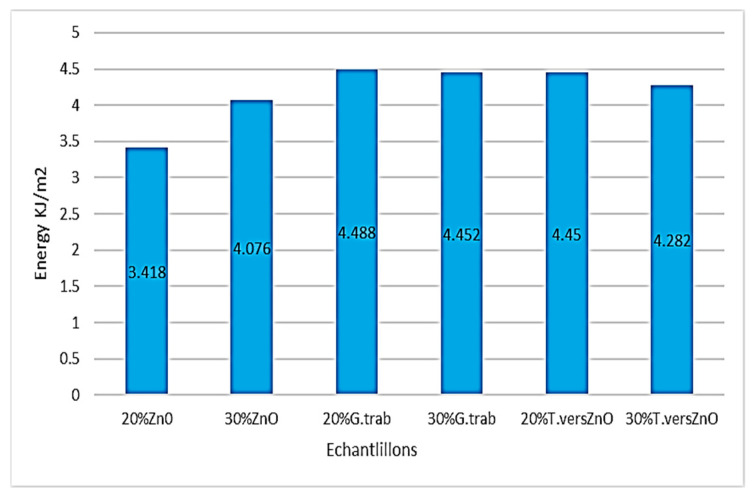
Izod impact energy of ZnO-treated HDPE/birch fiber composites exposed and non- exposed to rot. Legend: G.trab: exposed to *Gloephyllum trabeum*; T.vers: exposed to *Trametes versicolor*; ZnO: treated with zinc oxide; 20 and 30%: yellow birch fiber content.

**Figure 5 polymers-15-03664-f005:**
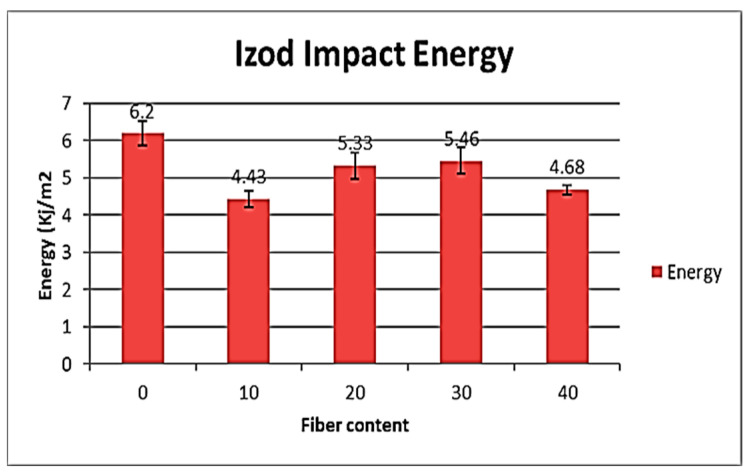
Impact energy of HDPE/yellow birch fiber [6].

**Figure 6 polymers-15-03664-f006:**
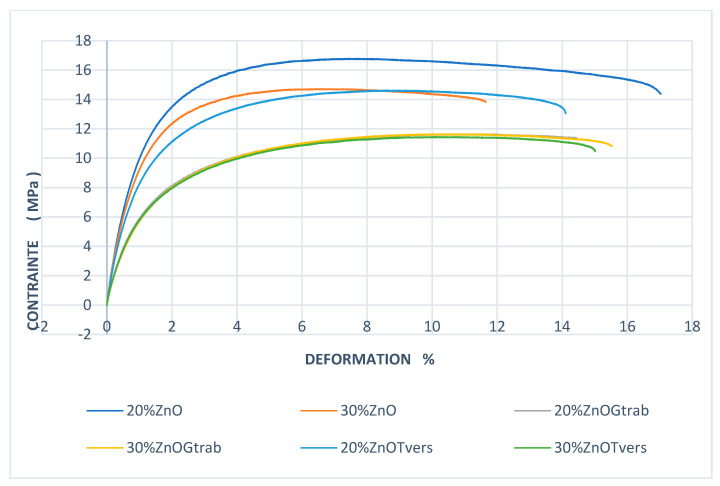
Stress-strain behavior of ZnO-treated samples exposed to rot. Legend: 20%ZnO: 20% samples treated with ZnO; 30%ZnO: 30% samples treated with ZnO; 20%ZnOGtrab: 20% samples treated with ZnO and exposed to *G. trabeum*; 30%ZnOGtrab: 30% samples treated with ZnO and exposed to *G. trabeum;* 20%ZnOTvers: 20% samples treated with ZnO and exposed to *T. versicolor*; 30%ZnOTvers: 30% samples treated with ZnO and exposed to *T. versicolor*.

**Figure 7 polymers-15-03664-f007:**
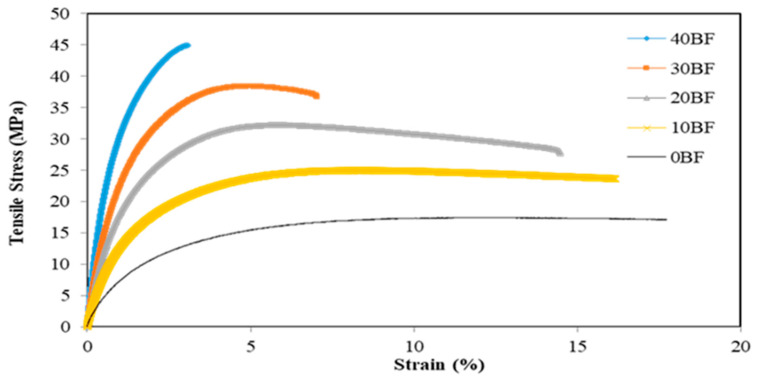
Stress–strain behavior of samples [6].

**Figure 8 polymers-15-03664-f008:**
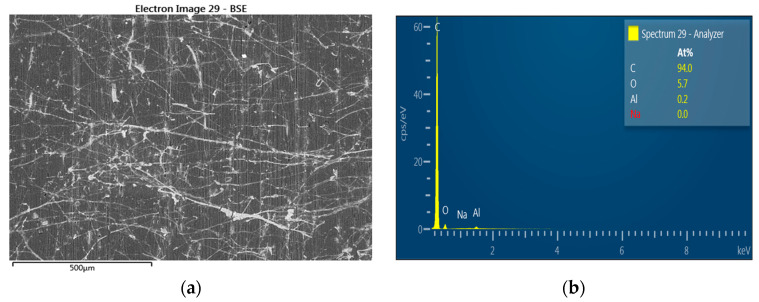
(**a**) Nontreated sample’s tangential section observed at G 100× *G. trabeum* hyphae are visible (**b**) EDX analysis.

**Figure 9 polymers-15-03664-f009:**
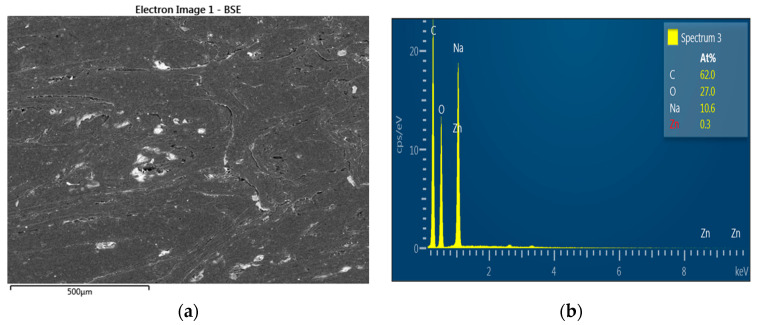
(**a**) ZnO-treated sample’s tangential section observed at G 100× exposed to *G. trabeum* and (**b**) EDX analysis.

**Table 1 polymers-15-03664-t001:** Evaluation of the biocomposite sample surfaces covered by the white- and brown-rot growth, together with the nomenclature of their classification.

Biocomposite Formulation	Main Surface Covered	
	White Rot	Brown Rot
20% yellow birch fibers	0	0
30% yellow birch fibers	0	0

Legend: 0 = no visible growth; 1 = up to 10%; 2 = 10–30%; 3 = 30–70%; 4 = > 70%.

**Table 2 polymers-15-03664-t002:** Tensile properties of treated/non-treated samples to rot.

Samples	Exposure to Rot	Young Modulus (GPa)	Strain (%)
Virgin HDPE 0% Yellow Birch Fiber (YBF)	Not exposed	1.51 ± 0.13	18.80 ± 1.64
	White	-	-
	Brown	-	-
20% YBF Biocomposite nontreated	Not exposed	2.67 ± 0.13	32.47 ± 0.19
	White	2.30 ± 0.02	28.27 ± 0.47
	Brown	2.36 ± 0.04	28.36 ± 0.25
20% YBF Biocomposite treated with ZnO	Not exposed	1.42 ± 0.05	16.58 ±2.18
	White	1.18 ± 0.026	11.42 ± 1.00
	Brown	1.17 ± 0.07	14.71 ± 0.83
30% YBF Biocomposite nontreated	Not exposed	3.37 ± 0.16	38.33 ± 0.47
	White	3.22 ± 0.06	35.15 ± 1.09
	Brown	3.42 ± 0.07	34.64 ± 1.40
30% YBF Biocomposite treated with ZnO	Not exposed	1.30 ± 0.04	10.93 ± 1.33
	White	0.79 ± 0.02	15.21 ± 1.08
	Brown	0.86 ± 0.01	16.68 ± 1.26

## Data Availability

The data presented in this study are available on request from the corresponding author.

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
