# Peer review of "ZnO Treatment on Mechanical Behavior of Polyethylene/Yellow Birch Fiber Composites When Exposed to Fungal Wood Rot"

_polymers, 2023, doi:10.3390/polym15183664_

Round 1

Reviewer 1 Report

The current work focuses on ZnO treatment on the mechanical behavior of polyethylene/yellow birch fiber composites when exposed to fungal wood rots. The author’s some effort into the manuscript, but major issues should be addressed. 

Abstract

- First, show the current manuscript's novelty and importance and then the main outputs.

-The first appearance of the abbreviation should have a full definition e.g. HDPE, G. trabeum, and T. versicolor.

Introduction 

-The introduction doesn’t provide sufficient background and all relevant references are not included.

e.g. no information on ZnO/HDPE or its properties, which is the important ingredients in the composites.

-The first appearance of the abbreviation should have a full definition e.g. WPC.

-The novelty of this work is not highlighted and the author's contribution was unclear compared to other previous works. 

Materials and Methods

- Materials and Methods section is well written, where provides sufficient details, and all information is included for easier reproduction by the reader:

Results

-The first appearance of the abbreviation should have a full definition e.g. YBF

-One of the main problems in the manuscript is that the authors show only results without interpretations or details of it. More details are required to explain the obtained results in each section.

e.g.Where is the control sample to compare and evaluate the obtained results of fungal growth on the Zinc oxide treated bio-composites made of HDPE and yellow birch fibers?

Why choose special 20 and 30% only? 

Why did the samples expose at day 1 of exposure change in color to samples with day 30 of exposure?

The importance and the role of ZnO in the composites are not discussed deeply. In addition, the effect of NaHO is not also discussed its influence.

-All figures need to redraw with high quality. 

-The thermal stability e.g TGA should be included to show the influence of ZnO on the stability of composites.

References

-Use only one style for all references.

Extensive editing of English language required

Author Response

suggestions have been considered and changes have been made in the manuscript.

control samples were those used in Koffi et al work. Indeed, he was the one that optimized and first to fabricate our samples by injection molding. his samples and ours had the same characteristics. (Koffi, A., Koffi, D. and Toubal, L., 2021. Mechanical properties and drop-weight impact performance of injection-molded HDPE/birch fiber composites. Polymer Testing, 93, p.106956)

in previous work, samples affectation by fungal attack was evidenced as it is shown in the attached document i sent you.

TGA analisis was performed but was not very concluding on ZnO presence. in fact mayor temperature decrease was due to the HDPE degradation at 500 degrees.(attached file)

we chose to produce 20 and 30% because most work confirmed fungal attack could be noticed only fro 50% (w/w) fiber content which was not the case in our study. despite low fiber content fungal attack had effect WPC performance. and we had to stop at 30% because our injection machine does not have much pressure and is the only one at our disposal.

Reviewer 2 Report

The publication good written. I have no comments or questions, I suggest to accept the publication.

Author Response

Thank you very much for reviewing the manuscript.

Reviewer 3 Report

The manuscript reports a study on the preparation of HDPE wood plastic composites treated with zinc oxide powder. The main idea is to utilise the antimicrobial properties of the ZnO to inhibit the growth of fungi (white rot and brown rot). Besides, the effect of addition of ZnO was also investigated in term of mechanical properties. The experimental works were properly done with clear objectives to be achieved. However, the manuscript is not properly written. It contains many badly constructed sentences and grammar mistakes, which requires further revision. Please find my comments and suggestions below:

- The camera pictures of of the chemicals, raw materials, specimens (before and after tested) in bulk form, machines and equipment are meaningless.

- Line 88: What is the meaning of "It was dissolved NaOH solution." & "The adequate solution was a molar"?

- Line 91: Any washing process of the fibers was conducted after the treatment process?

- Line 101: Procedure for preparing the bio-composite samples - the methodology is not clearly described. What is the meaning of "The remaining HDPE and the fibers were added and blended at 60 rpm and for 7 min". 

- Why 20 wt% and 30 wt% of fibers were used? As fas as biocomposites are concerned, normally higher fibers should be used (60-70 wt%). Please justify.

- Further details for parameters of granulation, injection molding. etc are needed.

- It is rather unclear why there is a need to dissolve ZnO in NaOH solution before the treatment of the fibers. The authors did not even mention sodium zincate which should be the main product of the reaction between ZnO and NaOH. Therefore, in my opinion, the term "ZnO treated biocomposites" is not suitable.

- The treatment of the fibers with NaOH does not seem to beneficial expect for the antifungal. Most mechanical strengths of the biocomposites are detrimentally affected by the ZnO treatment.

- The degree of treatment using ZnO and NaOH onto the fibers should be quantified. 

The manuscript is not yet ready to be published in the current form. In fact, additional experimental works are required to increase the impact and value of the work. Besides, in term of writing, the manuscript still requires very major revision in terms of grammar level, formatting errors, resolution quality of images, graph plotting, etc. 

The quality of English language of the manuscript is low. 

Author Response

Thank you very much for your suggestions, changes have been made on the manuscript.

no washing process was conduced, which was a mistake. The idea was to deeply treat the fiber for better homogeneity of ZnO.

the WPC methodology have been more explained now in manuscript. the HDPE is not introduced totally. A small part is poured first then maleic anhydride grafted to polyethylene is added; the remaining HDPE is added after that and then the fibers.

no washing process was realized after treatment which could be a mistake too.

indeeed some authors have affirmed that is only from 50% and more fiber content that WPC could be affected by fungal attack. Our research have shown us that even at 20% fiber content WPC mechanical performance is compromised. and on other hand the injection molding machine does not have much pressure for fabricating higher fiber content WPC, and is the only one on our disposal.

the idea was to use ZnO antifungal properties. Sodium zincate does not have known antifungal charatceristic as there is not much detailed information on its. Sodium zincate was an important product of the reaction but the main focus is on the ZnO itself.

the manuscript have been upgrated according to your suggestions also.

Reviewer 4 Report

1- The abstract must be rewritten, to include some of the important results of this research.

2-The novelty of this research is not clear.

3- Figure 1, should be moved to the discussion part.

4- What is the use of Figures 2, and 3 in the methodology? remove Figures 2, and 3 from the manuscript. Remove all the Figures from the methodology part.

5- Why Figure 11 is having a reference? It is not the result of the research?

6- Increase the resolution of the graphs in the discussion part.

7- All the graphs are drawn in very basic form and must redraw with a high resolution.

8- To improve the manuscript in the discussion and introduction part use the following references: https://doi.org/10.3390/polym15020272, and https://doi.org/10.1016/j.polymertesting.2020.106922

Minor editing of English language required.

Author Response

Thank you very much for your suggestions, manuscript have been rearranged accordingly.

meaningless figures have been removed and graph resolution improved.

formal figure 11 is referenced because there are the results of Koffi et al work (Koffi, A., Koffi, D. and Toubal, L., 2021. Mechanical properties and drop-weight impact performance of injection-molded HDPE/birch fiber composites. Polymer Testing, 93, p.106956.)   his results were our control samples , he is the one that optimized the injection machine parameter and realized studies on it. we used these parameters to fabricate our samples. ( Koffi, A., 2021. Étude des paramètres d'injection des composites de fibres naturelles et de l'amélioration des performances mécaniques du matériau pour l'impression 3D (Doctoral dissertation, Université du Québec à Trois-Rivières).

 i tried to check the references you suggested me but the page was not available.

Round 2

Reviewer 1 Report

All issues are solved and the manuscript is accepted in the present form.

Reviewer 3 Report

The authors have revised the manuscript and responded to the comment. 

Reviewer 4 Report

need to copy and paste the DOI into Google Scholar to use the suggested articles.